# Retinopathy of Prematurity—Targeting Hypoxic and Redox Signaling Pathways

**DOI:** 10.3390/antiox13020148

**Published:** 2024-01-25

**Authors:** Liyu Zhang, Francesco Buonfiglio, Achim Fieß, Norbert Pfeiffer, Adrian Gericke

**Affiliations:** Department of Ophthalmology, University Medical Center, Johannes Gutenberg University Mainz, Langenbeckstrasse 1, 55131 Mainz, Germany; lzhang03@uni-mainz.de (L.Z.); fbuonfig@uni-mainz.de (F.B.); achim.fiess@unimedizin-mainz.de (A.F.); norbert.pfeiffer@unimedizin-mainz.de (N.P.)

**Keywords:** retinopathy of prematurity, pathophysiology, signaling pathways, novel, molecular targets

## Abstract

Retinopathy of prematurity (ROP) is a proliferative vascular ailment affecting the retina. It is the main risk factor for visual impairment and blindness in infants and young children worldwide. If left undiagnosed and untreated, it can progress to retinal detachment and severe visual impairment. Geographical variations in ROP epidemiology have emerged over recent decades, attributable to differing levels of care provided to preterm infants across countries and regions. Our understanding of the causes of ROP, screening, diagnosis, treatment, and associated risk factors continues to advance. This review article aims to present the pathophysiological mechanisms of ROP, including its treatment. Specifically, it delves into the latest cutting-edge treatment approaches targeting hypoxia and redox signaling pathways for this condition.

## 1. Introduction

Retinopathy of prematurity (ROP) is a developmental vascular proliferative disease affecting the retina characterized by abnormal capillary growth in infants born preterm [1]. Key risk factors encompass low gestational age, low birth weight, and the utilization of prolonged mechanical, and particularly fluctuating, ventilation, often employed as a therapeutic measure in preterm infants [2,3,4]. The surge in premature births globally has led to a dramatically increased incidence of ROP. If left undetected and untreated, it can result in severe consequences for the visual system, including retinal detachment and irreversible vision impairment. ROP ranks among the primary causes of childhood blindness worldwide [5].

In the 1940s and 1950s, the unrestrained use of oxygen (O_2_) in industrialized nations sparked the initial epidemic of retinal disease in premature births, termed retrolental fibroplasia [6,7,8,9]. Subsequently, advancements in neonatal care and perinatal monitoring in the 1960s and 1970s increased the survival rate of extremely premature infants, causing a second surge in retinal disease [10,11]. This trend continued with a third peak affecting middle-income countries and regions like China, Southeast Asia, South Asia, South America, and Eastern Europe due to enhanced survival rates of very premature infants through improved neonatal care [12,13,14,15,16,17,18,19,20]. More recently, European countries and the US have witnessed a rising trend in ROP incidence. For instance, a study in the UK revealed a 4% incidence of ROP requiring treatment among preterm babies weighing < 1500 g [21]. Similarly, studies in Greece and Norway reported incidences of around 18.4% and 39.6%, respectively, among preterm infants [22]. In the US, the incidence rose from 11% in 2009 to 15% in 2018 among neonates meeting ROP screening criteria [23]. Currently, developed regions like the US and the UK tend to show a lower incidence, while developing regions such as India and Africa demonstrate slightly higher ROP incidence rates [24].

Despite advancements in ROP research and treatment, its incidence continues to rise, leading to escalated healthcare costs. Gyllensten et al. conducted a meta-analysis highlighting that the costs of ROP screening (ranging between USD 324–USD 1072 per child) and therapy (ranging between USD 38–USD 6500 per child) are considerably lower than the societal costs of resulting blindness (ranging between USD 26,686–USD 224,295), emphasizing the pivotal role of screening tools in combating this disorder [25]. Similarly, a cost-effectiveness analysis in Mexico and the US estimated substantial annual benefits of around USD 206 million and USD 205 million, respectively, by implementing national ROP screening and treatment programs [26]. Another systematic review reaffirmed the high cost-effectiveness of ROP screening and treatment in the UK, Canada, and the US [27].

Given the imperative to prevent ROP-associated blindness in children, ophthalmologists and public health experts must prioritize therapies and screening. Understanding the pathomechanisms of ROP is crucial for proposing innovative treatments targeting new pathways. This review aims to provide an extensive overview of advancements in understanding ROP’s etiopathogenesis, delving into the molecular cascades responsible for disease onset and progression. Furthermore, it discusses experimental studies testing novel treatment approaches targeting hypoxic and redox signaling pathways, paving the way for innovative therapeutic strategies to enhance ROP management.

## 2. General Aspects of Retinopathy of Prematurity

### 2.1. Classification, Diagnosis, and Risk Factors

The 2021 International Classification of Retinopathy of Prematurity (ICROP3) enables ophthalmologists to classify ROP through four fundamental parameters: zone, plus disease, stage, and extent [28]. Additionally, ICROP highlights the clinical significance of an aggressive ROP variant, aggressive retinopathy of prematurity (A-ROP), characterized by rapid pathologic neovascularization [28]. Retinal vascularization initiates at the 13th gestational week, completing at birth. The extent and location of retinal vascularization may mirror infant maturity and the risk of ROP development. The zones indicating vascularization status are [29,30]:Zone I: circle centered on the optic nerve head, having a radius equal to twice the distance between the optic nerve and the fovea.Zone II: circle centered on the optic nerve head, presenting a radius equal to the distance between the optic nerve and nasal ora serrata.Zone III: peripherical retinal area extending over Zone II.

Additionally, the terms “plus” and “pre-plus” disease are employed to characterize vascular abnormalities observed in ROP [31]. To elaborate, plus disease is characterized by discernible vascular changes in the posterior pole, manifesting as tortuous arterioles and dilated venules. On the other hand, pre-plus disease signifies noteworthy vascular changes that, while not as advanced as in plus disease, indicate abnormal development that may progress to the latter stage [28]. It is noteworthy that the assessment of plus disease has gained increasing significance as a prognostic factor, contributing to both diagnosis and determination of severity [31]. 

Beyond the differentiation between plus and pre-plus diseases, ROP is further classified into five stages. The first three stages (stage 1, stage 2, and stage 3) denote acute forms, and detailed presentations of these stages are provided below in Figure 1. The last two stages are characterized as partial and total retinal detachment, labeled stage 4 and stage 5, respectively. Disease extent is categorized using 30 sectors based on clock-hour positions [28]. Furthermore, clinical trials conducted by the Early Treatment for Retinopathy of Prematurity Cooperative Group have identified the advantages of managing ROP at threshold stages, leading to the classification of pre-threshold ROP into two types [32].

Type 1: high-risk pre-threshold ROP includes Zone 1 with + disease at any stage, Zone 1 stage 3 without + disease, or Zone 2 stage 2 or 3 with + disease, necessitating prompt therapy.Type 2: low-risk pre-threshold ROP comprises Zone 1 stage 1 or stage 2 without + disease, and Zone 2 stage 3 without + disease, recommended for follow-up.

Figure 1 illustrates the classification of ROP based on fundus zones, ROP stages, and types.

Various maternal prenatal conditions, including hypertensive disorders, gestational diabetes, maternal age, and smoking, are recognized factors associated with prematurity. Additionally, prenatal and perinatal factors such as assisted conception, mode of delivery, premature rupture of membranes, and chorioamnionitis contribute to the risk of preterm birth [5]. Notably, specific risk factors associated with ROP include prematurity itself, prolonged postnatal oxygen supplementation, apnea, sepsis, necrotizing enterocolitis, anemia, blood transfusions, and intraventricular hemorrhage [5,33,34,35]. Key predictors of risk are postulated to be low birth weight (BW), low gestational age (GA), and the rate of weight gain, with lower BW and GA posing a higher risk for severe ROP [36,37,38,39,40]. For example, the American Academy of Pediatrics (AAP) and the American Academy of Ophthalmology (AAO) recommend screening for infants born at or before 30 weeks GA or those with a BW less than 1500 g [41]. However, universal screening guidelines are lacking, as the Royal College of Paediatrics and Child Health (RCPCH) suggests screening for infants born under 32 weeks GA or those weighing less than 1501 g. In Canada, screening is recommended for any infant born at or before 30 weeks GA, regardless of birth weight, and for infants with a BW of ≤1250 g [42,43]. Moreover, different birth weight percentiles, such as small-for-gestational-age (below the 10th percentile), appropriate-for-gestational-age, or large-for-gestational-age, need to be considered during screening, as they correlate with diverse risk probabilities for developing ROP [44].

Infant factors including ethnicity, gender, multiple births, Apgar score, postnatal weight gain, insulin-like growth factor I (IGF-1) levels, hyperglycemia, insulin levels, comorbidities, and treatments such as pulmonary complications, anemia, transfusion, erythropoietin (EPO), and thrombocytopenia also play a role in ROP pathogenesis [5].

### 2.2. Screening and Diagnostic Tools

Guidelines for ROP screening in preterm infants vary across countries [45,46,47,48,49,50,51,52,53]. While ROP screening is crucial for mitigating the visual complications in preterm infants, it presents drawbacks such as elevating scores in assessment tools like “Crying, Requires O_2_, Increased vital signs, Expression, and Sleeplessness” (CRIES) and the “Premature Infant Pain Profile” (PIPP) [54]. Additionally, the use of mydriatic agents like phenylephrine and cyclopentolate during ROP screening can trigger tachycardia and systemic hypertension in preterm infants [55]. Hence, there is a pressing need to refine ROP screening methods to minimize these undesirable effects.

In recent years, concerted scientific efforts have aimed to develop evidence-based ROP screening algorithms to identify preterm infants at a high risk of developing ROP. For instance, recognizing the crucial role of Insulin-like Growth Factor-1 (IGF-1) in retinal blood vessel growth, Lofqvist et al. proposed the WINROP online prediction model. This algorithm relies on low postnatal weight gain as an indirect indicator of slower serum IGF-1 increase and impaired retinal vascular growth, aiding in identifying infants at risk of ROP requiring treatment [56]. By integrating birth weight, gestational age, and weekly weight measurements, this model estimates the risk of severe ROP development [57].

Several studies evaluating the WINROP algorithm showcased high sensitivity (around 90–100%) but relatively lower specificity (around 30–50%) [58,59,60,61,62,63].

An alternative predictive model, the Children’s Hospital of Philadelphia (CHOP) ROP model, developed by Binenbaum and colleagues, employs three parameters—birth weight, gestational age, and daily weight gain rate—to assess severe ROP risk [64]. Subsequently, the same research group further refined this with the Postnatal Growth and ROP (G-ROP) criteria. This advancement aimed to reduce the number of infants requiring examinations and demonstrated increased sensitivity and specificity, particularly for type 1 ROP, compared with currently recommended guidelines [65,66]. These evolving algorithms hold promise for enhancing ROP screening by offering more accurate risk assessments and potentially minimizing unnecessary interventions.

Interestingly, a recent investigation led by Bortea and colleagues unveiled a significant association between the development of ROP in extremely premature and very premature neonates and inflammatory markers [67]. Specifically, the study, encompassing neonates, extremely premature infants (GA < 28 weeks), and very premature infants (GA between 28 and 32 weeks), revealed markedly higher levels of interleukin (IL)-6 and lactate dehydrogenase levels at birth and three days postnatally in the extremely premature group compared with the other groups. Conversely, C-reactive protein (CRP) levels at three days were higher in the very premature infants’ group. Additionally, a positive correlation was established between umbilical cord inflammation and ROP severity. Furthermore, elevated CRP and IL-6 levels were linked to an increased risk of developing ROP stage 2 or above, underscoring their potential as biomarkers for predicting ROP risk [67]. In summary, this investigation highlights the significance of assessing inflammatory markers in predicting the risk of ROP in extremely premature and very premature infants.

### 2.3. Natural Course, Long-Term Sequelae, and Prognosis

The natural regression of stage 1 ROP occurs in approximately 90% of cases, but as the stages progress, the regression rates decline significantly to about 50–60% in stage 2 and roughly 6% in stage 3 [68]. Advancement to stages 4 and 5, with retinal detachment, leads to irreversible visual loss. The CRYO-ROP study indicated that therapy significantly reduces the risk of adverse visual outcomes from 52% to 30% over a 15-year follow-up period, displaying improved visual acuity outcomes in treated patients at 3, 10, and 15 years [69]. Recent studies of our own revealed that extremely preterm adults, with or without postnatal ROP, tend to undergo more frequent ophthalmological check-ups throughout their lives, likely due to long-term complications in eye development and subsequent retinal disorders [70,71]. In the Gutenberg Prematurity Eye Study, individuals with postnatal ROP and particularly participants who needed treatment had a higher risk of impaired stereopsis, amblyopia, and reduced vision-related quality of life in adulthood [72]. Moreover, preterm delivery was shown to have little effect on absolute refractive error but was associated with anisometropia in adulthood. Notably, ROP treatment using cryo- and laser coagulation was shown to increase the risk of refractive error, lens opacification, and impaired accommodation [73]. In addition, corneal morphology was reported to be influenced by gestational age and birth weight percentile, while anterior chamber depth and lens thickness were shown to be affected by ROP treatment [74]. Additionally, a thicker foveal and a higher prevalence of foveal hypoplasia, potentially affecting visual acuity, were observed in cases of ROP requiring treatment compared with less severe ROP or preterm individuals without ROP. Moreover, individuals born extremely preterm or those with untreated ROP showed an increased vertical cup-to-disc ratio in adulthood, while all people born preterm revealed thinner peripapillary retinal nerve fiber layer thickness compared with individuals born full-term [75,76]. This might predispose these individuals to degenerative optic neuropathies [75,77]. These results collectively suggest that extreme preterm birth and postnatal ROP occurrence may lead to various anomalies in retinal development, impacting visual acuity in adulthood.

Despite advancements in treatment options, at least 50,000 children worldwide suffer blindness due to ROP annually, with approximately 600 preterm infants affected in the US each year [78]. The occurrence of ROP-related blindness varies by country and is influenced by regional socioeconomic development [79]. In high-income countries, ROP-related blindness is relatively rare, constituting less than 10% of cases of irreversible visual loss [80,81]. Conversely, middle- and low-income countries, constrained by limited public health resources, may exhibit higher rates of ROP-related blindness, accounting for up to 40% of blindness cases in these regions [16,79].

## 3. Insights into the Pathophysiology of Retinopathy of Prematurity

### 3.1. Retinal Development and Disease Pathogenesis

During physiological retinal angiogenesis, blood vessels begin to form around the 14–15th week of gestation, originating from the optic nerve head and expanding centrifugally towards the retinal periphery [82]. By 36 weeks of gestation, the nasal portion of the retina becomes vascularized, while the temporal area completes this process by the 40th week. Consequently, preterm infants exhibit incompletely vascularized retinas, with the extent of the avascular zone contingent upon their gestational age [83].

Under normal circumstances, the hypoxic conditions typical of the intrauterine environment stimulate retinal vascularization by prompting the expression of hypoxia-inducible factor 1α (HIF-1α). This factor regulates the expression of various oxygen-sensing genes, including crucial proangiogenic factors like vascular endothelial growth factor (VEGF) [84]. Although the VEGF family encompasses several members, including VEGF-A, -B, -C, -D, and placental growth factor (PlGF), it is VEGF-A that predominantly drives retinal angiogenesis [85,86].

VEGF, primarily released by neuroglia, initiates retinal blood vessel formation through the migration of vascular endothelial cells in a paracrine manner [87]. As retinal angiogenesis progresses, hypoxic conditions diminish, leading to the cessation of HIF-1 activation and its target genes. However, premature exposure to oxygen (hyperoxia) in an immature retina significantly suppresses HIF-1 and VEGF activity, resulting in oxidative stress and the emergence of avascular retinal regions [82]. Nitro-oxidative stress, characterized by an imbalance between the abundant generation of reactive oxygen species (ROS) and reactive nitrogen species (RNS) and antioxidative defense mechanisms, leads to excess ROS and RNS. This imbalance triggers dramatic structural molecular changes and the activation of inflammatory and cell death pathways, a condition detectable in various eye diseases [88,89]. It is noteworthy that infants possess reduced antioxidant defenses [90]. In this context, Buhimish and colleagues reported that preterm births exhibit a lack of compensatory upregulation of nonenzymatic antioxidant reserves, such as glutathione (GSH) and plasma total free radical-trapping antioxidant potential [91]. Given the diminished antioxidative capability and increased vulnerability to oxidative stress in preterm neonates, it is not surprising that several oxidative biomarkers, such as malondialdehyde (MDA), 8-hydroxy 2-deoxyguanosine (8-OHdG), and the GSH/GSSG ratio, have been discussed as potential diagnostic tools for ROP [92].

An overabundance of ROS is described during the first phase of ROP, a status of hyperoxia that manifests as vaso-obliteration, characterized by decreased levels of HIF-1α, VEGF, and IGF-1 [93]. Subsequently, an ischemic phase ensues, gradually progressing into a proliferative stage marked by abnormal and dysfunctional neoangiogenesis. This phase ultimately results in intravitreal fibrosis, retinal traction, and detachment [82,93].

Ashton et al. introduced the two-phase hypothesis on ROP pathogenesis, demonstrating that exposing healthy cats to 70–80% oxygen for four days induces newly formed capillaries, leading to a process of “vaso-obliteration.” Upon returning to normal air exposure, a phase of “vasoproliferation” is observed [94]. In the initial phase (phase I), physiological retinal angiogenesis is delayed due to high oxygen exposure, resulting in vascular occlusion, reduced serum IGF-1, and delayed expression of VEGF receptors 2 [95]. Subsequently (phase II), retinal and vitreous neoangiogenesis occurs alongside increased levels of HIF-1α, VEGF, IGF-1, placental growth factor, erythropoietin (EPO), metalloproteinase (MMP)-2, MMP-9, and angiopoietin (Ang)-2 [96]. Another model, the oxygen-induced retinopathy (OIR) mouse model, mimics high oxygen levels similar to Ashton’s experiments. In this model, exposure to constant high oxygen (75% O2) causes newly formed capillaries to regress, leading to central areas of vaso-obliteration. Upon returning to room air, relative hypoxia triggers the release of angiogenic factors, promoting the vasoproliferation of blood vessels into the vitreous [97].

Figure 2 illustrates Phases 1 and 2 in the pathogenesis of ROP, focusing on the distinct expression of key mediators contributing to the onset of the disease.

### 3.2. Exploring Molecular Cascades in Retinopathy of Prematurity

Throughout the various pathological stages of ROP, a multitude of molecular signaling pathways emerge, contributing to inflammatory processes and an abundance of ROS and RNS—two crucial factors in the early stages of ROP [83,98].

Subsequent sections delve into the primary pathways responsible for vaso-obliteration and vaso-proliferation during ROP pathogenesis.

#### 3.2.1. The Central Role of Nitro-Oxidative Stress and Inflammatory Factors

In ROP, endothelial cell apoptosis triggered by oxidative stress is implicated in the process of vaso-obliteration that occurs in the retina during the initial phase of the disorder. Gu et al. demonstrated in bovine retinal endothelial cells that hyperoxia-induced nitro-oxidative stress leads to retinal capillary endothelial cell apoptosis, potentially by impeding growth factor-induced activation of the PI3K/Akt signaling pathway [99]. The pro-oxidative enzyme nicotinamide adenine dinucleotide phosphate (NADPH) oxidase (NOX), present in isoforms NOX1, NOX2, and NOX4, generates high levels of superoxide (O_2_∙^−^), one of the most detrimental ROS, and has been associated with oxidative stress and vascular neoangiogenesis in OIR models [100,101,102]. Wang et al. reported in a rodent model of OIR that NOX4 might regulate intravitreal neovascularization mediated by VEGFR2 via activated signal transducer and activator of transcription (STAT) 3 within endothelial cells [101]. Studies by Saito et al. on animal models of ROP showcased that hyperoxia activates NOX, leading to an excess of ROS, culminating in apoptosis and neoangiogenesis independently of VEGF [103,104]. Furthermore, Byfield et al. demonstrated in a rodent model subjected to repeated oxygen fluctuations that hyperoxia-induced NOX activation leads to intravitreal VEGF-associated vascularization through a Janus tyrosine kinase (JAK)2/STAT3 pathway, and inhibiting this signaling pathway reduced neoangiogenesis [105]. However, VEGF signaling pathways seem to play a role in both phases of ROP, influencing pro-inflammatory and pro-oxidative processes. The same research group later revealed that VEGF-induced STAT3 activation blocked retinal angiogenesis by downregulating local expression of erythropoietin (EPO) in Müller cells during phase 1 of OIR [106]. Ren et al.’s investigation in a rodent model exposed to hyperoxia demonstrated that hyperoxia-induced STAT3 signaling enhances hepcidin expression—a key protein involved in iron balance—suggesting a potential compensatory mechanism to counteract iron overload linked to neoangiogenesis and proposing targeting molecules to regulate iron regulation [107].

Nitric oxide (NO) is pivotal in regulating vasodilation processes. Three main isoforms of NO synthases (NOS)—neural (n)NOS, inducible (i)NOS, and endothelial (e)NOS—are described in the literature. To counteract vaso-obliteration, vasodilation via NO synthesis occurs during the early phases of ROP [108]. However, differential NO responses are associated with the redox state [109]. As the disease progresses, excess NO synthesis becomes detrimental, favoring neoangiogenesis [110]. Importantly, normal NOS activity relies on the availability of the cofactor (6R)-5,6,7,8-tetrahydrobiopterin (BH_4_). Edgar et al. assessed in a murine model of OIR that exposure to oxygen led to decreased BH_4_ levels in the retinas, lungs, and aortas of mice, resulting in increased NOS-related ROS [111]. In circumstances of reduced BH_4_, uncoupled eNOS activity leads to nitro-oxidative stress in ROP pathogenesis. Specifically, in hyperoxia, impaired eNOS generates peroxynitrite (ONOO^−^) rather than physiological NO. Alongside eNOS, iNOS has also been reported as a pathogenetic factor in ROP. For instance, hypoxia-induced activation of iNOS in a murine model of OIR was linked to HIF-1α activation, VEGF expression, and PI3K/Akt signaling during neoangiogenesis, and inhibiting iNOS reduced the expression of these mediators [112]. However, NO also plays a significant role in neoangiogenesis processes during the vaso-proliferative phase, being fundamental in events of vascular permeability and leakage observed in retinopathies, affecting adherent junctions and endothelial cell polarity [113,114].

Notably, nitro-oxidative stress can interfere with prostanoid metabolism, exacerbating vaso-obliteration and contributing to avascular retinal onset. Reactive nitrogen species were observed in a murine OIR model to promote an isomerization of arachidonic acid to trans-arachidonic acid, involved in upregulating the anti-angiogenic factor thrombospondin-1 [115]. In this context, the role of phospholipase A2 (PLA-2) becomes noteworthy—an enzyme triggered by hypoxia and ROS abundance, influencing prostanoid metabolism via arachidonic acid release [116]. This molecule acts as a substrate for cyclooxygenase (COX), which converts it into proangiogenic eicosanoids, such as prostaglandins, prostacyclin, and thromboxane. Barnett et al. showed that suppressing PLA-2 decreased proangiogenic prostaglandins and intravitreal neoangiogenesis in an OIR model [116].

Arginase, present in isoforms arginase 1 and 2, hydrolyzes L-arginine to ornithine and urea, also producing glutamate [117]. This enzyme has been implicated in neural regeneration and protection [118]. Preterm infants exhibit low arginine levels, a crucial amino acid for retinal vascular development [119,120]. Arginine administration, particularly intravitreally with glutamine, countered neoangiogenesis in an OIR mouse model [121]. Importantly, arginase functionality and expression are heightened in contexts of inflammation and ROS excess, potentially interfering with NOS activity by competing for the substrate L-arginine. This competition indirectly triggers NOS uncoupling, leading to an overproduction of ONOO^−^ [122,123]. Arginase 1 has been associated with neuroprotection [124], while arginase 2 might play a role in the events leading to retinal injuries, closely linked with nitro-oxidative stress and inflammation [125]. Shosha et al. proposed that NOX2-related O_2_∙^−^ induces an upregulation of arginase 2 in ischemia/reperfusion injury, contributing to neurovascular degeneration [126].

In addition to oxidative stress, inflammatory events play a crucial role in the pathogenesis of ROP [127]. Cytokines such as IL-1β, tumor necrosis factor-α (TNF-α), and IL-6 are identified as primary drivers of inflammation, capable of inducing the overexpression of various inflammatory mediators, including chemokines and adhesion molecules. Specifically, within the hypoxic neonatal retina, retinal microglia can produce substantial amounts of IL-1β and TNF-α, ultimately promoting the death of retinal ganglion cells [128]. Furthermore, a study by Rivera and colleagues demonstrated in an OIR model that IL-1β is associated with retinal microvascular degeneration, triggering the release of the proapoptotic/repulsive factor semaphorin-3A from neurons [129]. Subsequent investigations by the same research group on the same model revealed the early pivotal role of IL-1β in the choroid, contributing to the involution of choroidal blood vessels and causing a loss of retinal pigment epithelium and photoreceptors [130]. As a consequence of cytokine downstream activation, chemokines also play a role in the pathogenesis of ROP, facilitating chemotaxis and the recruitment of immune cells to sites of inflammation. Specifically, chemokines implicated in ROP include IL-8, “RANTES” (Regulated and Normal T-cell Expressed and Secreted), and monocyte chemotactic protein 1 [131,132,133,134,135,136]. Taken together, inflammatory factors are pivotal in the pathophysiology of ROP, considering their role in orchestrating, together with oxidative stress, an amplification of the aberrant immune-mediated activation that leads to retinal cell death and choroidal degeneration.

#### 3.2.2. The Crucial Involvement of HIF-1α and VEGF

HIF is a transcription factor composed of two subunits: HIF-1α (or its analogs HIF-2α and HIF-3α) and HIF-1β [137]. In normoxic conditions, HIF-1α undergoes hydroxylation by a prolyl hydroxylase domain (PHD) in the cytosol [138]. However, under hypoxic conditions, the enzymatic activities of PHD are inhibited, resulting in an increase in HIF-1α expression. Subsequently, HIF-1α binds to HIF-1β in the nucleus, forming the HIF-1 complex, which activates angiogenic mechanisms to help cells adapt to hypoxia. HIF-1α orchestrates the expression of several neoangiogenic mediators, including VEGF, EPO, angiopoietin (Ang)-1, and Ang-2, all observed to be upregulated in phase 2 of ROP [139]. Notably, under hypoxic conditions, HIF-1α is pivotal in reprogramming cellular metabolism, enhancing glycolysis, and increasing mitochondrial NADPH synthesis [140]. Even under normoxic conditions, HIF-1α activity can be induced by ROS and stabilized by inflammatory cytokines and growth factors like IGF-1 and TGF-β [141,142]. Enhanced HIF-1α activity due to relative intrauterine hypoxia is critical for physiological retinal vascular development [143]. However, in premature births, postnatal hyperoxia suppresses HIF-1α activity, leading to reduced VEGF release and consequent retinal capillary obliteration [144,145]. Stabilizing HIF-1α might represent a potential molecular target to halt the progression to phase 2 of ROP, as discussed in Section 4.2.4.

As mentioned earlier, VEGF plays a crucial role in both phases of ROP. Reduced VEGF levels under hyperoxia contribute to vaso-obliteration via endothelial cell apoptosis [146,147]. Conversely, in phase 2 under hypoxia, increased retinal VEGF levels act on endothelial cells through a paracrine route [148]. VEGF’s signaling involves the downregulation of retinal EPO in Müller cells via STAT3 activation. Furthermore, retinal endothelium expresses VEGFR-2, a receptor pivotal in neoangiogenic events and responsible for directing dividing endothelial cells in the developing retina [149]. Upregulation of VEGFR-2 disrupts dividing endothelial cells, potentially driving them to develop outside the retina, as observed in models of intravitreal neovascularization [150]. Inhibiting VEGFR-2 has shown promise in reducing intravitreal neoangiogenesis in preclinical investigations using the OIR model [151]. In Section 4.2.4, we delve into the primary experimental strategies targeting the VEGF/VEGFR axis in ROP.

Figure 3 provides a schematic overview illustrating the primary molecular pathways that unfold during hyperoxia and hypoxia in ROP.

## 4. Treatment Intervention in Retinopathy of Prematurity

### 4.1. Established Therapy Options

The primary aim of managing ROP is to prevent vision loss and safeguard retinal structures. Cryotherapy emerged in the 1980s as a routine treatment, preventing fibrovascular retinal detachment by eliminating the avascular retina and halting abnormal angiogenesis in the vitreous [152]. Dedicated clinical trials have demonstrated the benefits of cryotherapy in improving eye anatomy and visual development [153]. However, studies employing indirect laser delivery systems for the eye suggested that cryotherapy outcomes were less favorable compared with laser treatment, potentially resulting in more severe myopia [154].

Laser photocoagulation therapy has notably reduced the progression of ROP to retinal detachment [155]. This approach targets the peripheral retina, aiming to decrease the risk of further angiogenesis and disease recurrence. Laser photocoagulation offers the benefit of a one-session therapy with a long-lasting effect. However, it is not without drawbacks. There is a possibility of skip areas following laser treatment, leading to disease reactivation [156]. Furthermore, both cryotherapy and laser photocoagulation may induce long-term sequelae, resulting in structural abnormalities and functional deficits such as reduced visual acuity, diminished visual fields, and significant myopia [156]. Additionally, laser therapy carries significant adverse effects, including the development of cataracts, anterior segment ischemia, and glaucoma [73,74,157,158,159]. Furthermore, although it is a rare complication, exudative retinal detachment has also been documented as a potential consequence of laser photocoagulation, a procedure employed in the treatment of infants affected by ROP [160].

Biologics represent the new pharmacological frontier for ROP, with current use including bevacizumab, aflibercept, and pegaptanib, albeit off-label, while ranibizumab has been the first licensed drug for ROP in Europe [156]. Notably, the long-term ocular and systemic effects of this therapy remain unclear [161]. The two most commonly used biologics for ROP are bevacizumab and ranibizumab [162]. Bevacizumab, the initial anti-VEGF agent used for ROP treatment, is a humanized monoclonal antibody that blocks all VEGF isoforms [163,164]. Findings from the Bevacizumab Eliminates the Angiogenic Threat (BEAT)-ROP study indicate a significantly lower recurrence rate of zone I ROP with intravitreal bevacizumab treatment compared with conventional laser treatment [165]. Unlike traditional laser treatment, intravitreal bevacizumab allows for continued blood vessel growth [166]. Ranibizumab, a monoclonal antibody fragment targeting VEGF-A, neutralizes VEGF-A to restrict the proliferation of abnormal retinal vasculature. The RAINBOW trial, involving 87 neonates across 26 countries, demonstrated higher treatment success rates at 24 weeks with intravitreal ranibizumab (80% of patients) compared with the laser treatment group (66%) in ROP infants [162]. Importantly, ranibizumab treatment did not induce systemic VEGF inhibition, positioning it as a promising alternative to ROP laser therapy.

Crucially, the benefits of anti-VEGF therapy lie in its relative speed and simplicity of administration, as well as a rapid therapy response. Biologics can protect the peripheral visual field and cause less myopia. Consequently, these drugs are specifically indicated in cases of zone I ROP and aggressive posterior ROP [156]. According to a report by the American Academy of Ophthalmology, intravitreal anti-VEGF intervention proves as effective as laser photocoagulation in inducing regression of acute ROP [167]. However, disadvantages include a higher recurrence of ROP, necessitating prolonged follow-up to monitor incomplete retinal vascularization [167]. Moreover, anti-VEGF therapy leads to a reduction in systemic VEGF levels in preterm neonates, and the impact on the developing organ systems of premature neonates remains substantially unknown. Therefore, the use of biologics for ROP is cautiously recommended, with an awareness of potential unknown adverse effects.

### 4.2. Exploring Emerging Molecular Targets

#### 4.2.1. Exploring Antioxidant Strategies

Vitamin C, vitamin E, and lutein play pivotal roles as components in antioxidant defense mechanisms [168]. These substances have been investigated as potential strategies for antioxidant treatment in ROP. Vitamin C acts as a free radical scavenger and contributes to the regeneration of the antioxidant form of vitamin E. Despite promising results in preclinical studies, clinical trials have not consistently demonstrated significant advantages in managing ROP through vitamin C supplementation [92,169]. Clinical studies on vitamin E supplementation have yielded conflicting findings regarding its impact on ROP progression and the occurrence of adverse effects [170]. While preclinical studies have shown decreased avascular zones under hyperoxia [171], dedicated meta-analyses have reported benefits alongside increased rates of sepsis and necrotizing enterocolitis [172,173]. A more recent clinical study on vitamin E supplementation did not report associations with adverse effects but confirmed benefits in reducing ROP rates [174].

Lutein, recognized as a safe antioxidant agent and used in managing eye diseases such as age-related macular degeneration, has been tested in a murine model of OIR. In this model, lutein demonstrated the ability to diminish vascular leakage and promote normal endothelial tip cell formation, contributing to retinal vascular development [171]. Omega-3 long-chain polyunsaturated fatty acids, known for their effectiveness in countering various neurodegenerative disorders and suppressing apoptosis through the reduction of oxidative stress [171], have shown promise in a recent clinical trial. The trial reported that enteral lipid supplementation with docosahexaenoic acid and arachidonic acid, primary polyunsaturated fatty acids (PUFAs), effectively reduced the risk of severe ROP by 50% for neonates born at less than 28 weeks gestational age [175]. In summary, the effects of potential antioxidant supplements on managing ROP are subject to ongoing discussion, and further research is warranted to elucidate their efficacy and potential benefits.

#### 4.2.2. Targeting Nuclear Factor-Erythroid 2-Related Factor 2 (Nrf2) for ROP Management

The transcription factor nuclear factor-erythroid 2-related factor 2 (Nrf2) plays a crucial role in modulating the expression of essential antioxidative agents, providing fundamental protection against ROS [176]. Uno and colleagues investigated the impact of hyperoxia in Nrf2 knockout compared with wild-type mice, revealing a beneficial effect of Nrf2 in the retina exposed to hyperoxia-related oxidative stress, crucial for promoting vascular endothelial cell survival [177]. Notably, the authors also noted that the protective role of Nrf2 during normal retinal development aligns with a specific window of time, coinciding with angiogenesis. This suggests that precise intervention timing, such as the use of Nrf2 modulators, may be critical in the treatment of ROP [177]. Consistent with these findings, Deliyanti et al. explored the effect of a Nrf2 activator, dh404, in a murine model of OIR. They observed a reduction in ROS levels, suppression of vaso-obliteration in phase I, and mitigation of neovascularization, vascular leakage, and inflammation in phase II [178]. In contrast, Liang and Wang investigated the impact of brusatol, a naturally occurring Nrf2 inhibitor extracted from *Brucea javanica*, in a rodent model of OIR. They found that brusatol could mitigate retinal microglial activation, neovascularization, and inflammation. This included a downregulation of VEGFR1, VEGFR2, TNF-α, and iNOS [179].

The existing literature suggests that the effect of modulating Nrf2 in ROP models is still not fully understood. While Nrf2 activation enhances the antioxidant response, its downregulation may also be beneficial by decreasing angiogenic and inflammatory agents regulated by Nrf2. Considering the interconnected and overlapping events of inflammation, neoangiogenesis, and oxidative stress in ROP, the timing of administering Nrf2 modulators may be essential to achieve the desired beneficial effects in counteracting ROP development.

#### 4.2.3. Targeting the STAT3 Signaling Pathway in ROP Management

Activation of the STAT3 signaling pathway represents a potential convergence point between two pivotal pathogenetic cascades in ROP: oxidative stress and the VEGF signaling pathway. Therefore, exploring molecules with the capability to suppress this pathway becomes of particular interest. In a murine model of OIR, Bartoli and colleagues demonstrated that the administration of fluvastatin, a 3-hydroxy-3-methylglutaryl-Coenzyme A (HMG-CoA) reductase inhibitor, preserved retinal neovascularization. This effect was achieved through the prevention of the upregulation of key factors such as VEGF, HIF-1α, phosphorylated STAT3, and ICAM-1. Additionally, fluvastatin exhibited antioxidant effects by reducing O_2_∙^−^ formation and lipid peroxidation in the ischemic retina [180]. In a more recent preclinical study, Chen et al. shed light on the role of the C-CBL protein, an E3 ubiquitin-protein ligase, in a murine model of OIR. They emphasized its crucial function in negatively controlling JAK2/STAT3/VEGF-related angiogenesis, highlighting that its upregulation suppresses neoangiogenesis in the retinas of OIR mice [181]. Ren and colleagues, in a rodent model of ROP, demonstrated that S3I-201, a STAT3 pathway inhibitor, downregulated the expression of STAT3 and VEGF mRNA levels, effectively countering neoangiogenesis in ROP [107]. In another investigation, the E3 ubiquitin ligase synoviolin (SYVN1) was examined for its potential in ROP management. SYVN1, known for its physiological functions in recognizing misfolded proteins in endoplasmic reticulum-associated degradation (ERAD), was upregulated through adenoviral vectors in a mouse model of OIR. This led to the ubiquitination and degradation of STAT3, reduced levels of phosphorylated STAT3, and a decrease in the release of VEGF. Collectively, these actions effectively countered neovascularization in the context of ROP [182].

#### 4.2.4. Targeting HIF-1α and VEGF

Jiang and colleagues demonstrated that inhibition of HIF-1α suppresses the production of pro-angiogenic factors that cause the neovascular phase [183]. Usui-Ouchi and associates demonstrated that intravitreal injection of peptides derived from the intrinsically disordered protein CITED2, an endogenous negative feedback regulator of HIF-1α prevented ROP in a mouse model of OIR [184]. Huang et al. recently showed in an animal model of OIR that recombinant thrombomodulin domain 1 (rTMD1) significantly decreased retinal neovascularization and contributed to normal physiological vessel growth through inhibition of the HIF-1α-VEGF pathway [185]. Zhao and coworkers demonstrated in a murine model of OIR that celastrol, a naturally occurring molecule extracted from *Tripterygium wilfordii* Hook F., has been assessed to be capable of mitigating retinal neovascularization via inhibition of miR-17-5p/HIF-1α/VEGF signaling pathway, thereby being suggested as a therapeutic candidate for prevention and treatment in ROP [186].

Caffeine, a common treatment for premature infants with apnea, has shown potential for preventing ROP. Research by Aranda et al. revealed that caffeine could prevent ROP by upregulating genes associated with the sonic hedgehog signaling pathways, involving VEGF and IGF-1, and contributing to neuroprotection and angiogenesis [187]. Additionally, a meta-analysis indicated that early administration of caffeine reduces the likelihood of requiring laser photocoagulation therapy for managing ROP [188].

Vitamin A, a crucial fat-soluble organic compound in eye homeostasis [189], has been explored in systemic retinoic acid administration during hyperoxia, showing an increase in VEGF expression [190]. Studies have suggested that vitamin A reduces retinal angiogenesis in animal models of OIR by inhibiting VEGF production, potentially impacting the progression and incidence of ROP [191,192,193].

A compelling new molecule currently under investigation for its anti-inflammatory properties in the context of ROP is stanniocalcin-1. Dalvin and colleagues conducted studies using rodent models of OIR by comparing knockout mice for stanniocalcin-1 with wild-type controls. Their findings revealed a more severe establishment of ROP, characterized by increased avascular retina and vaso-proliferative areas in the absence of stanniocalcin-1. Notably, when stanniocalcin-1 was expressed, it demonstrated the capability to reduce the levels of VEGF-A. This observation suggests that stanniocalcin-1 holds promise as a potential molecular target for mitigating the progression of ROP [194].

#### 4.2.5. The Role of Steroids in Managing ROP

Within the realm of steroidal drugs, triamcinolone stands out as a molecule under investigation in preclinical studies for its potential to counteract inflammatory events in ROP. For example, intravitreal injection of triamcinolone acetonide in a neonatal rat model of OIR has shown efficacy in reducing neovascularization by decreasing IGF-1 receptor phosphorylation [195]. In a more recent investigation, Öhnell et al. compared the incidence of type 2 ROP versus type 1 ROP among infants treated with dexamethasone eye drops and those untreated. They observed a treatment frequency of around 74% in infants with type 2 ROP who did not receive dexamethasone before laser ablation, compared with 24% in infants with type 2 ROP who received dexamethasone eye drops for type 2 ROP. The study concluded that such eye drops could substantially prevent these infants from developing type 1 ROP [196]. Nevertheless, the use of steroids in ROP remains a subject of debate. Prenatal exposure to dexamethasone has been associated with significantly lower odds of stage 2 or higher ROP [197]. However, the timing of postnatal corticosteroid administration is crucial. Early postpartum use can be beneficial in reducing ROP incidence, while late administration (more than seven days after birth) increases the risk of severe ROP [198,199]. Long-term corticosteroid use in ROP patients can also elevate the risk of progressing to severe stages [200]. A recent retrospective cohort study by Shekhawat and colleagues on 1695 infants with gestational age ≤ 32 weeks and/or birth weight ≤ 1500 g demonstrated that cumulative dose and duration of postnatal steroid use were independently associated with the severity of ROP and peripheral avascular retina [201].

In summary, based on the existing literature, the use of steroids in the context of ROP needs to be approached with great prudence and caution.

#### 4.2.6. Exploring Matrix Metalloproteinases (MMPs) in ROP Treatment

Matrix Metalloproteinases (MMPs) constitute a group of endopeptidases critical for the breakdown of components within the extracellular matrix, with activation triggered by factors such as ROS, low pH, and heat treatment [202,203,204]. In response to angiogenic stimuli such as VEGF, fibronectin, and TNF-α, retinal pigment epithelial cells exhibit an increased secretion of MMP-2 and MMP-9 [205]. Das et al. demonstrated that heightened expression of MMP-9 contributes to elevated VEGF levels and retinal neovascularization during phase 2 of retinopathy in an OIR model. Significantly, intravitreal injection of TIMP-1, an inhibitor of MMP, successfully prevented the expression levels of MMP-9, VEGF, and retinal neovascularization. This underscores the potential efficacy of MMP-9-targeted interventions in the treatment of ROP [206]. More recently, Patnaik and colleagues unveiled that the decreased level of opticin, an anti-angiogenic factor, is induced through the release of MMP-9 by activated microglia under hypoxia in the vitreous of ROP eyes. They demonstrated that the use of doxycycline and EDTA to suppress MMP-9 can rescue the expression of opticin, potentially countering vaso-proliferative processes [207].

#### 4.2.7. Exploring Potential β-Adrenoceptor Targets: Focus on Propranolol

Propranolol, a non-selective beta-adrenergic blocker commonly employed in pediatric care for conditions such as arterial hypertension and arrhythmias, demonstrates generally good tolerance in humans [208,209]. There is growing interest in the potential use of prophylactic oral beta-blockers, such as propranolol, to limit the progression to stage 3 ROP and reduce the need for anti-VEGF drugs or laser therapy [210]. This effect is attributed to evidence suggesting that β_2_-adrenoceptors play a role in upregulating vascular endothelial growth factor VEGF and IGF-1 levels, making them relevant in the pathogenesis of various neovascular retinal diseases [211,212].

Studies on preterm newborns with stage 2 ROP who received oral propranolol have indicated a reduction in the risk of advancing to stage 3 or 4. However, it is crucial to note that some preterm infants may experience severe adverse reactions, such as hypotension and bradycardia, particularly in the presence of sepsis episodes, anesthesia induction, or tracheal irritation [213]. To mitigate these risks, alternative routes of propranolol administration have been proposed. In a multicenter clinical trial, the topical application of 0.2% propranolol eye drops in newborns at stage 1 ROP significantly reduced disease progression to stage 2 or 3 plus [214]. Although promising for preventing severe ROP, further randomized controlled studies are essential to assess the long-term effects, determine optimal dosages, and establish the duration of therapy for propranolol treatment.

#### 4.2.8. Targeting Succinate and Adenosine Pathways

During episodes of ischemic hypoxia, oxidative metabolites such as succinate serve as signaling factors that respond to compromised energy states, leading to an increase in retinal vascularization [215]. Similarly, hypoxia induces the accumulation of purine products (ATP, ADP, AMP, and adenosine) and activates purinergic receptors, further promoting neovascularization [216,217]. Elevated levels of succinate and adenosine during hypoxia activate their cognate G protein-coupled receptors (GPCRs), restoring adequate blood flow by dilating blood vessels and stimulating angiogenesis [218]. In an OIR model, succinate and adenosine contribute to the proliferative phase of ROP by modulating the expression or activity of their respective GPCRs. Furthermore, the down-regulation of GPR91 disrupts normal retinal vascular development and reduces abnormal intravitreal neovascularization [219]. Antagonists targeting the A2B adenosine receptor have shown efficacy in reducing pre-retinal neovascularization [216]. GPR91 plays a crucial role in preventing excessive growth factor secretion and decreasing intravitreal neovascularization [219,220], making it a promising target for the development of molecular interventions [221].

## 5. Concluding Remarks

As our understanding of ROP has evolved since the initial discovery of retrolental fibroplasia, advancements across technology, scientific research, and clinical medicine have profoundly influenced our comprehension of its pathogenesis. Technological strides in regulating the environmental conditions for preterm infants have notably enhanced the survival rates of extremely premature infants.

Scientific and technological progress in unraveling the molecular mechanisms underlying pathophysiology has shed light on both normal and aberrant developmental angiogenesis. Exploring neurovascular interactions and their role in cognitive function and angiogenesis represents an emerging frontier. Significant clinical trials have contributed crucial insights into oxygenation, screening, classification, and treatment modalities for severe ROP. There is a growing awareness that ROP manifests variably across different regions globally, prompting the consideration of tailored approaches to screening and treatment. Despite these strides, the treatment of ROP remains an active area of research. There is a critical need for targeted therapies that mitigate abnormal vascular proliferation while fostering the physiological development of the retinal vasculature and safeguarding the delicate health of the developing infant. Of note, numerous treatment approaches targeting hypoxia and redox signaling pathways in the context of ROP have shown promising results in preclinical studies. Hence, more clinical studies are warranted in this field.

## Figures and Tables

**Figure 1 antioxidants-13-00148-f001:**
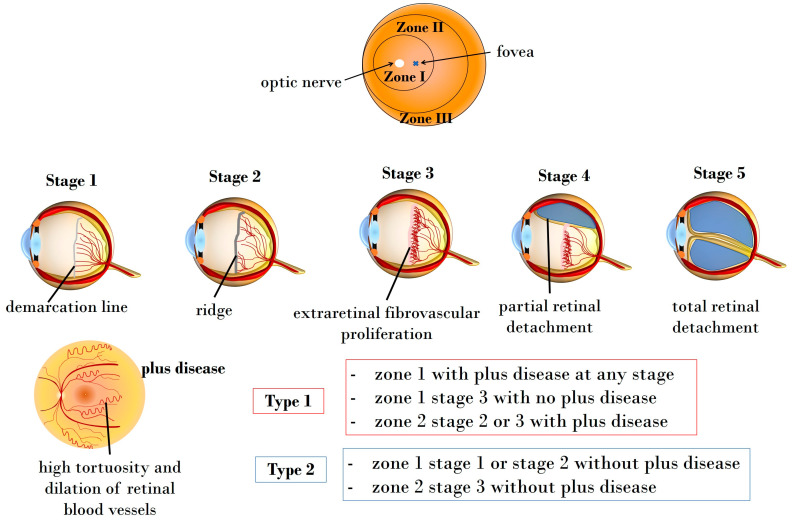
Illustration depicting the zones, stages, and types of ROP. Zone I encompasses the circular area centered on the optic nerve head, having a radius equal to twice the distance between the optic nerve and the fovea, while Zone II extends as a circle centered on the optic nerve head, presenting a radius equal to the distance between the optic nerve and nasal ora serrata. Zone III covers the peripherical retinal area, extending over Zone II. Stage 1 manifests as a demarcation line, delineating the boundary between the physiologically vascularized retina and the peripheral avascular retina. In Stage 2, this line progresses into a distinct ridge. Stage 3 marks the onset of extraretinal neovascularization and hemorrhages. Stage 4 indicates partial and stage 5 total retinal detachment, respectively. The plus disease is characterized by pronounced vascular dilation and tortuosity.

**Figure 2 antioxidants-13-00148-f002:**
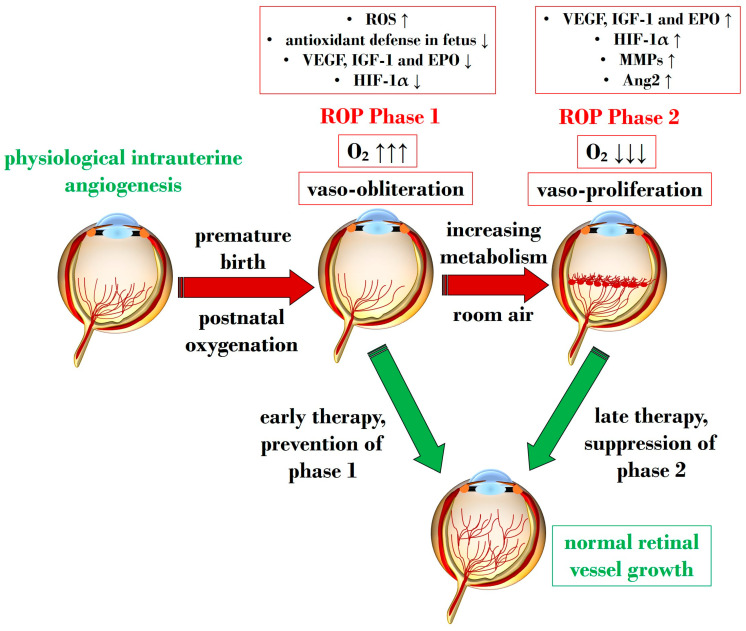
Schematic representation depicting the two pathogenetic phases in ROP, highlighting distinct expression levels of key mediators in the retina. ROP: retinopathy of prematurity; ROS: reactive oxygen species; O_2_: oxygen; VEGF: vascular endothelial growth factor; IGF-1: insulin-like growth factor 1; EPO: erythropoietin; HIF-1α: hypoxia-inducible factor 1 alpha; MMP: metalloproteinase; Ang-2: angiopoietin 2. Upward black arrows indicate upregulation or increased concentration, downward black arrows indicate downregulation or decreased concentration.

**Figure 3 antioxidants-13-00148-f003:**
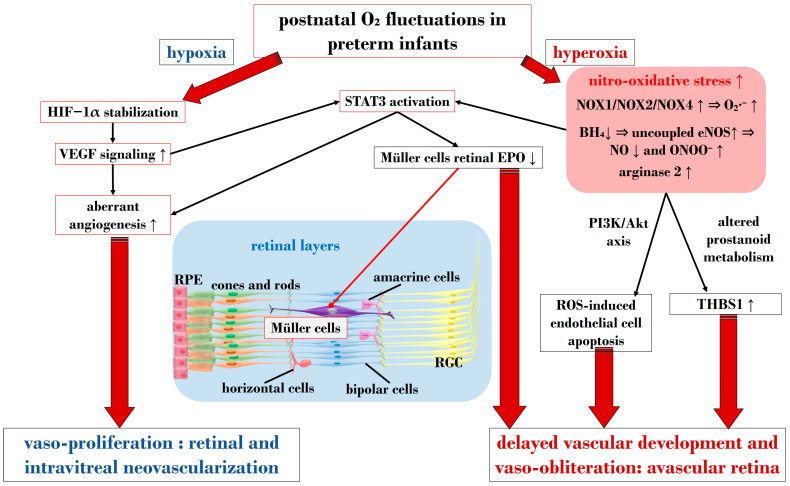
Schematic overview depicting the primary molecular pathways activated during phases 1 and 2 in ROP. ROS: reactive oxygen species; NOX: nicotinamide adenine dinucleotide phosphate oxidase; O_2_: oxygen; O_2_∙^−^: superoxide; ONOO^−^: peroxynitrite; VEGF: vascular endothelial growth factor; EPO: erythropoietin; HIF-1α: hypoxia-inducible factor 1 alpha; BH_4_: tetrahydrobiopterin; NO: nitric oxide; eNOS: endothelial nitric oxide synthase; STAT3: signal transducer and activator of transcription 3; RGC: retinal ganglion cell; RPE: retinal pigment epithelium; THBS1: thrombospondin 1. Upward black arrows indicate upregulation or increased concentration, downward black arrows indicate downregulation or decreased concentration.

## Data Availability

Not applicable.

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
