# Peer review of "Retinopathy of Prematurity—Targeting Hypoxic and Redox Signaling Pathways"

_antioxidants, 2024, doi:10.3390/antiox13020148_

Round 1

Reviewer 1 Report

Comments and Suggestions for Authors

The study under review meticulously examines various publications addressing the pathogenesis and therapeutic approaches for retinopathy of prematurity, addressing the significant challenges associated with this medical condition. The paper is thoughtfully organized, utilizing tables and figures effectively to enhance data comprehension. The conclusions drawn are clear, well-supported by the results, and presented in high-quality scientific English.

A minor comment from this reviewer notes a slight overemphasis on self-citation in the bibliography.

Nevertheless, overall, this well-written manuscript is poised to make a substantial impact on the readers of this journal and can serve as a valuable resource in the field of research on retinopathy of prematurity.

Author Response

We appreciate the reviewer's positive comments. However, we would like to address one concern related to the perceived overemphasis on self-citations in the bibliography. It is important to note that while we referenced only 10 of our own publications on ROP or oxidative stress in the retina, our collaborative efforts have contributed to over 100 studies on these topics. We believe this level of self-citation is appropriate, especially considering the manuscript includes a total of 221 citations.

Reviewer 2 Report

Comments and Suggestions for Authors

See attached file

Author Response

We greatly appreciate the reviewer for providing such positive feedback.

Reviewer 3 Report

Comments and Suggestions for Authors

I would congratulate the authors to an excellent overview regarding the main goal : Author: This review article aims to present the pathophysiological mechanisms of ROP, including its treatment.

This review give an proper and a scientific basis for current  knowledge in this field ( Classification, Diagnosis; Screening and Diagnostic Tools; Natural Course, Long-Term Sequelae, and Prognosis: Insights into the Pathophysiology of Retinopathy of Prematurity; Treatment Intervention in Retinopathy of Prematurity) I am  sorry to say that in my opinion this manuscript missing  more detail information about recent development in  the field of Inflammatory and steroids in ROP regarding to  pathophysiology and treatment. Below some recent publications.

1.A Prospective Analysis of the Retinopathy of Prematurity Correlated with the Inflammatory Status of the Extremely Premature and Very Premature Neonates. Borțea CI, Enatescu I, Dima M, Pantea M, Iacob ER, Dumitru C, Popescu A, Stoica F, Heredea RE, Iacob D. Diagnostics (Basel). 2023 Jun 18;13(12):2105.

2.Impact of postnatal steroids on peripheral avascular retina and severity of retinopathy of prematurity. Shekhawat PS, Ali MAM, Kannekanti N, Koechley H, Mhanna C, Pinto M, Farghaly MAA, Mhanna M, Aly HZ, Sears JE. Pediatr Res. 2023 Dec;94(6):1966-1972. doi: 10.1038/s41390-023-02673-4. Epub 2023 Jun 8

3.Dexamethasone Eye Drops for the Treatment of Retinopathy of Prematurity. Öhnell HM, Andreasson S, Gränse L. Ophthalmol Retina. 2022 Feb;6(2):181-182.

Öhnell HM, Andreasson S, Gränse L. Dexamethasone Eye Drops for the Treatment of Retinopathy of Prematurity. Ophthalmol Retina. 2022 Feb;6(2):181-182.

Rivera JC, Holm M, Austeng D, Morken TS, Zhou TE, Beaudry-Richard A, Sierra EM, Dammann O, Chemtob S. Retinopathy of prematurity: inflammation, choroidal degeneration, and novel promising therapeutic strategies. J Neuroinflammation. 2017 Aug 22;14(1):165

4. Exudative Retinal Detachment Following Laser Photocoagulation for Retinopathy of Prematurity: A Rare Complication. Moinuddin O, Bonaffini S, Besirli CG. Ophthalmic Surg Lasers Imaging Retina. 2019 Apr 1;50(4):242-246

As you know there have been some review with or without metanalysis in this field in recent years and tuhs review is important, but as inflammatory factors seems be a new and important tool in evaluation  and in treatment of these children (recently even at AAO 2023), my suggestion  would be that this chapter should be further described in this review. 

Author Response

We sincerely appreciate the reviewer for providing these insightful comments and suggestions. Following these recommendations, we have incorporated additional content referencing the corresponding articles in the following sections:

  • Chapter 2: Diagnostic, Subheader 2.2. (lines 168–180, added sentences are underlined).

  • Chapter 3: Pathophysiology, Subheader 3.2.1. (lines 355–373, added sentences are underlined).

  • Chapter 4: Treatment, Subheaders 4.1. and 4.2.5. (lines 435–438 and 580–602, added sentences are underlined).

Round 2

Reviewer 3 Report

Comments and Suggestions for Authors

From my point of view, it´s an important review